# Changes in Photosystem II Complex and Physiological Activities in Pea and Maize Plants in Response to Salt Stress

**DOI:** 10.3390/plants13071025

**Published:** 2024-04-03

**Authors:** Martin A. Stefanov, Georgi D. Rashkov, Preslava B. Borisova, Emilia L. Apostolova

**Affiliations:** Institute of Biophysics and Biomedical Engineering, Bulgarian Academy of Sciences, Acad. G. Bonchev Str., Bl. 21, 1113 Sofia, Bulgaria; martin_12.1989@abv.bg (M.A.S.); megajorko@abv.bg (G.D.R.); preslavab12345@gmail.com (P.B.B.)

**Keywords:** antiradical activity, antioxidant activity, LHCII, low-temperature chlorophyll fluorescence, oxygen-evolving complex, pigment composition, Q_A_ reoxidation

## Abstract

Salt stress significantly impacts the functions of the photosynthetic apparatus, with varying degrees of damage to its components. Photosystem II (PSII) is more sensitive to environmental stresses, including salinity, than photosystem I (PSI). This study investigated the effects of different salinity levels (0 to 200 mM NaCl) on the PSII complex in isolated thylakoid membranes from hydroponically grown pea (*Pisum sativum* L.) and maize (*Zea mays* L.) plants treated with NaCl for 5 days. The data revealed that salt stress inhibits the photochemical activity of PSII (H_2_O → BQ), affecting the energy transfer between the pigment–protein complexes of PSII (as indicated by the fluorescence emission ratio F_695_/F_685_), Q_A_ reoxidation, and the function of the oxygen-evolving complex (OEC). These processes were more significantly affected in pea than in maize under salinity. Analysis of the oxygen evolution curves after flashes and continuous illumination showed a stronger influence on the PSIIα than PSIIβ centers. The inhibition of oxygen evolution was associated with an increase in misses (α), double hits (β), and blocked centers (S_B_) and a decrease in the rate constant of turnover of PSII reaction centers (K_D_). Salinity had different effects on the two pathways of Q_A_ reoxidation in maize and pea. In maize, the electron flow from Q_A_- to plastoquinone was dominant after treatment with higher NaCl concentrations (150 mM and 200 mM), while in pea, the electron recombination on Q_A_Q_B_- with oxidized S_2_ (or S_3_) of the OEC was more pronounced. Analysis of the 77 K fluorescence emission spectra revealed changes in the ratio of the light-harvesting complex of PSII (LHCII) monomers and trimers to LHCII aggregates after salt treatment. There was also a decrease in pigment composition and an increase in oxidative stress markers, membrane injury index, antioxidant activity (FRAP assay), and antiradical activity (DPPH assay). These effects were more pronounced in pea than in maize after treatment with higher NaCl concentrations (150 mM–200 mM). This study provides insights into how salinity influences the processes in the donor and acceptor sides of PSII in plants with different salt sensitivity.

## 1. Introduction

Climate changes over the past decade have impacted soil salinization and agricultural production [1]. The adverse effects of salt stress on plants result in limitations to their growth and development [2]. It has been demonstrated that photosynthesis, one of the key processes in plants, is significantly affected by salinity [3]. The impact on the photosynthesis is induced by osmotic stress and ion-specific toxicity [4,5]. High salt concentrations lead to the disruption of thylakoid membrane organization and a decrease in photosynthetic efficiency [6,7,8,9]. Salt stress triggers an overproduction of reactive oxygen species (ROS), such as singlet oxygen, superoxide radicals, hydrogen peroxide, and other free oxygen radicals [10,11,12,13]. The ROS damage proteins, lipids, nucleic acid, and other macromolecules [14,15,16,17]. The production and scavenging of ROS are crucial for plant responses to adverse environmental conditions [18].

The activity of ROS leads to chlorophyll degradation and membrane lipid peroxidation, altering membrane fluidity [13]. Some authors suggest that the decrease in chlorophyll, the increase in lipid peroxidation, and the amount of H_2_O_2_ can serve as markers indicating oxidative damage [13]. Salt stress has varying effects on pigment composition across different plant species. A decrease in chlorophyll and carotenoid contents was observed in salt-sensitive species, while in salt-tolerant species, an increase in these pigment levels was registered under high-salinity conditions [3,19].

Previous studies have demonstrated that salinity influences the protein level in thylakoid membranes [20,21,22]. When examining particles of photosystem II (PSII) from spinach, it was found that the dissociation of the extrinsic proteins of the oxygen-evolving complex (OEC) occurs at a high salt content [23]. It has also been shown that the amounts of D2 and Chl a/b binding protein (CP 29) of PSII vary depending on the degree of salinity [24]. A study of cucumber revealed a decrease in the proteins of the light-harvesting complex of PSII (LHCII) and D2. There was also a decrease in the lipids in the thylakoid membrane as well as an increase in the level of saturated fatty acids under high salinity [25]. All these structural changes influence the function of the photosynthetic apparatus. Recent studies revealed a salt-induced increase in the energy transfer from PSII to PSI (photosystem I) and changes in the energy transfer between the pigment–protein complexes of PSII [26,27]. Under salt stress, the inhibition of the functions of both photosystems is observed, but the effect is stronger on PSII than PSI [28,29,30]. The degree of injury caused by salt is influenced by several factors, including the specific type and concentration of salt as well as the duration of exposure [30,31]. The impact of oxidative stress depends on the balance between the generation and removal of ROS [32]. Previous studies have demonstrated that the antioxidant enzyme systems and non-enzymatic antioxidants like alpha-tocopherols and flavonoids protect against oxidative damage induced by salinity stress [33]. It has been shown that flavonoids prevent lipid peroxidation under stressful conditions [34,35].

In the present study, the effect of different concentrations of NaCl on two important crop plants, pea (*Pisum sativum* L.) and maize (*Zea mays* L.), was investigated. Our recent study revealed a different salt sensitivity of these plant species [36]. We hypothesize that a detailed study of the energy transfers and functions of the donor and acceptor sides of the PSII complex will more clearly show how changes in these processes induced by salt stress are related to the tolerance of plant species to salinity. We studied the energy transfer among pigment–protein complexes, the kinetic parameters of oxygen evolution, Q_A_ reoxidation pathways, and the photochemical inhibition of PSII. The changes in the pigment composition, markers of oxidative stress, total antioxidant and antiradical activity, and membrane injury were also studied. The experimental results provide new information about the influence of salinity on the donor and acceptor side of the PSII complex in pea and maize.

## 2. Results

### 2.1. Pigment Composition

The salt treatment influenced the chlorophyll (Chl) and carotenoid (Car) content in both studied plants (Table 1), as changes were observed in plants after applying 150 mM and 200 mM NaCl. The treatment of the studied plants with 150 mM NaCl and 200 mM NaCl led to a larger decrease in the Chl amount in pea (from 29% to 58%) than in maize (from 14% to 41%). The reduction in the Car content was also more pronounced in pea (from 26% and 52%) than in maize (from 12% to 32%) (Table 1). The data also revealed an increase in the Chl a/b ratio only in pea after treatment with higher NaCl concentrations (150 mM and 200 mM) (Table 1). At the same time, an increase in the Car/Chl ratio was registered in both studied plant species after treatment with 200 mM NaCl (Table 1).

### 2.2. Oxidative Stress Markers

The application of NaCl (50–200 mM NaCl) leads to an increase of the oxidative stress markers (MDA and H_2_O_2_) in the studied plant species (Figure 1). According to the experimental data, there was a greater increase in both oxidative markers in pea compared to maize. Following the treatment of samples with the highest NaCl concentration (200 mM), the content of MDA increased by 134% in pea and by 117% in maize, while the H_2_O_2_ content increased by 299% and 110% in pea and maize, respectively.

### 2.3. Antioxidant Activities and Total Flavonoids

The free radical scavenging capacity (DPPH) and ferric-reducing antioxidant power (FRAP) were used to assess the antioxidant activity in the studied plant species (Figure 2). The FRAP and DPPH activities increased after treatment with 150 mM and 200 mM NaCl in pea, while in maize, they increased only after applying 200 mM NaCl. The increase in DPPH values was larger in pea (by 77%–132%) than in maize (16%). The FRAP activities increased in pea by 33% and 60% after treatment with 150 mM and 200 mM NaCl, respectively, while in maize, they increased only at 200 mM NaCl by 18% (Figure 2). The data also revealed that flavonoids were increased after treatment with 150 mM and 200 mM NaCl in pea (by 11–13%) and in maize (by 26–32%) (Figure 3).

### 2.4. Membrane Injury Index

The membrane injury index (MII) was used to assess the membrane damage after treatment with different NaCl concentrations (Figure 4). This index was enhanced after salt treatment in both studied plant species. The application of the highest NaCl concentration resulted in a larger increase of the MII in pea (35%) than in maize (14%).

### 2.5. Low-Temperature Chlorophyll Fluorescence

Chlorophyll fluorescence emission spectra at low temperature (77 K) were used to assess the energy transfer between the pigment–protein complexes of the photosynthetic apparatus [26,37]. The spectra of the thylakoid membranes of maize and pea exhibited three maxima (685 nm, 695 nm, and 735 nm).

The fluorescence ratio F_735_/F_685_ characterizes the energy transfer from PSII to PSI, whereas the ratio F_685_/F_695_ characterizes the energy between the pigment–protein complexes within PSII. The ratio F_735_/F_685_ increased in pea after treatment with 150 mM and 200 mM NaCl, whereas in maize only the highest concentration (200 mM NaCl) influenced this ratio. In addition, the F_685_/F_695_ ratio decreased in pea exposed to 150 and 200 mM NaCl, while it was unchanged in maize (Table 2).

The decomposition of the low-temperature chlorophyll emission spectra provides information about the contribution of pigment–protein complexes in thylakoid membranes to the total 77 K chlorophyll emission spectrum [38]. The main bands in all variants of studied plant species had maxima at 680, 685, 695, 700, 720, and 735 nm, correspondingly assigned to LHCII (trimers and monomers, LHCII^T+M^), the reaction center of the PSII complex (PSII RC), the primary antenna complex of PSII (PSII antenna), LHCII (aggregated trimers, LHCII^A^), the core complex of PSI (PSI core), and the antenna complex of PSI (PSI antenna). A decrease in the fluorescence from LHCII^T+M^ was observed in thylakoid membranes of pea after treatment with 150 mM and 200 mM NaCl, while in maize it was observed after treatment with the highest NaCl concentrations (Table 3). At the same time, a decrease in the PSII RC was observed after treatment with higher NaCl concentrations (150 mM and 200 mM), as the effect was stronger in pea (27–36%) than in maize (9–10%). Data also revealed an increase of PSI antenna in pea after treatment with 150 mM and 200 mM NaCl, while in maize only after treatment with 200 mM NaCl (Table 3).

### 2.6. Photochemical Activity of PSII

The electron transport mediated by PSII in the presence of the electron acceptor BQ (H_2_O → BQ) was established to assess the photochemical activity of PSII [26,39]. The results showed inhibition of PSII-mediated electron transport in the two studied species after treatment with NaCl (Figure 5). Inhibition of this process was observed in pea after applying all studied NaCl concentrations (50 mM–200 mM), while in maize, it was observed after the addition of higher concentrations (150 mM and 200 mM). Data also revealed the degree of inhibition in the two species was different, i.e., the inhibition in pea was greater than in maize (at 150 mM NaCl with 20% and at 200 mM with 34%) (Figure 5).

### 2.7. Decay Kinetics of the Flash-Induced Variable Fluorescence

The dark relaxation of chlorophyll fluorescence excited by a single saturating light characterizes Q_A_ reoxidation [40]. The fluorescence signal was fitted with two components (fast A_1_ and slow A_2_) with times t_1_ and t_2_, which characterize two pathways of Q_A_ reoxidation [35]. The data also revealed that the impact of the high salt concentrations (150 mM and 200 mM NaCl) on the ratio A_1_/A_2_ in the studied species was different, i.e., this ratio increased in maize and decreased in pea (Figure 6). In addition, the data revealed that after applying the highest NaCl concentration, the component A_1_ decreased in pea by 56% and increased in maize by 14% (Appendix A). The time t_1_ increased by 30% in maize and by about three-fold in pea. An increase of time t_2_ was also registered in pea (Appendix A).

### 2.8. Oxygen Evolution under Flash and Continuous Illumination

The data showed that NaCl treatment led to the decrease of the oxygen evolution under continuous illumination (A) (Figure 7). This parameter decreased more in pea (62%) than in maize (16%) after treatment with the highest NaCl concentration (200 mM) (Figure 7). The amplitudes of the decay of the oxygen burst were fitted with two components (fast, A_F_ and slow, A_S_) with rate constants k_F_ and k_S_. It has been proposed that the components A_F_ and A_S_ correspond to the PSIIα and PSIIβ centers located in the grana and stroma lamellae, respectively [41]. At all studied NaCl concentrations, the A_F_/A_S_ ratio decreased only in pea, while in maize this ratio was unchanged. The evaluation of salt-induced alterations in the kinetic characteristics of the two PSII populations (PSIIα and PSIIβ) in maize and pea was conducted by investigating the rate constants, characterizing PSIIα (k_F_) and PSIIβ (k_S_). The data also revealed that the values of k_F_ and k_S_ decreased in pea between 28% and 52% after treatment with higher salt concentrations (150 mM and 200 mM NaCl), while in maize, significant changes in the rate constants were not observed (Figure 7).

The parameter Y_3_, characterizing flash oxygen evolution, was influenced more strongly than parameter A after treatment with 150 mM and 200 mM NaCl (Figure 7 and Figure 8). The decrease in this parameter (Y_3_) was by 80% in pea and by 24% in maize after treatment with the highest NaCl concentration (200 mM) (Figure 8).

The more detailed information about the impact of NaCl on the PSII complex gives the kinetic parameters of the oxygen evolution: the percent of PSII centers in the most reduced state in dark conditions (S_0_), misses (α) and double hits (β), the blocked PSII reaction centers (S_B_), and the rate constant of turnover of oxygen-evolving centers for the release of one O_2_ molecule (K_D_). The parameters So, α, and β increased after treatment with 150 mM and 200 mM NaCl, as the effects were more pronounced in pea than in maize. The data also revealed an increase in the blocked oxygen-evolving PSII centers (S_B_) (Table 4). The increase of the blocked reaction centers after treatment with 200 mM NaCl in pea was 98% and in maize 33%. In addition, salt treatment influenced to a much greater extent the constant K_D_ in pea compared to maize. This constant decreased in pea after treatment with 150 mM NaCl and 200 mM NaCl in the range of 22%–42%, while in maize a slight decrease of 7% was found only at the highest concentration of NaCl (200 mM) (Table 4).

### 2.9. Principal Component Analysis

Principal component analysis (Appendix A and Appendix A) showed that the first two components explain 96.59% of the variability. Maize and pea treated with 200 mM NaCl showed a negative correlation with the rate constants (k_F_, k_S_), the ratio of fast to slow components of oxygen evolution under continuous illumination (A_F_/A_S_), the rate constant of turnover of PSII reaction centers (K_D_), and 77 K chlorophyll fluorescence ratios F_685_/F_695_. On the other hand, a positive correlation was established between salt-treated plant variants and the time of the fast component of the dark relaxation of chlorophyll fluorescence excited by a single saturating light (t_1_), the blocked PSII reaction centers (S_B_) and the 77 K chlorophyll fluorescence ratio F_735_/F_685_. The most significant changes between the control and NaCl-treated plants occurred in the parameters S_B_, A_F_/A_S_, F_735_/F_685_, and k_F_. In addition, very big changes were observed in the pea treated with 200 mM NaCl compared to the other investigated variants.

## 3. Discussion

Salinity is a major environmental factor that strongly impacts photosynthetic machinery. Prior studies have demonstrated that elevated salinity results in disorganization of the grana thylakoids and affects the organization and functionality of the photosynthetic complexes [6,7,8,9,42]. Under salt stress, the inhibition of PSII is more pronounced than that of PSI [29,43]. Additionally, it has been observed that both the donor and acceptor side of the PSII complex are influenced [44]. This study provides new, comprehensive insights into the effects of salinity on the processes in the PSII complex.

The experimental results revealed that higher NaCl (150 mM and 200 mM) concentrations in both studied species decrease the Chl content differently (Table 1). A similar reduction in the Chl amount was registered in chickpea, *Solanum lycopersicum*, *Triticum aestivum*, *Ricinus communi*, and other plant species [3,45,46,47,48]. A salt-induced reduction in the chlorophyll content could be attributed to impaired chlorophyll biosynthesis and/or increased chlorophyll degradation, although the impact of these processes varies in plant species [49]. The decrease in Chl content in pea corresponds with an increase of the Chl a/b ratio (Table 1), suggesting a reduction of the LHCII and a decrease of the number of granal thylakoids [50,51,52], i.e., having an influence on the organization of the thylakoid membranes. Consistent with our suggestion, electron microscopic studies show that high salinity (100 mM and 200 mM) alters the structure of the chloroplasts and causes significant disintegration of thylakoids [42]. Salt stress led to decreased Car content in both species after applying higher NaCl concentrations (150 mM and 200 mM), with the Car reduction less pronounced in maize than in pea. Considering the crucial role of the Car as an effective antioxidant responsible for ROS quenching and photoprotection of photosynthesis [53], it could be suggested that this is one of the reasons for higher salt sensitivity in pea than in maize.

The salinity enhanced ROS production; however, plants have different adaptive mechanisms to mitigate the negative effect of oxidative stress [32,33]. Data in this study revealed increased levels of H_2_O_2_ and MDA as well as enhanced antioxidant power (FRAP activity) and radical scavenging activity (DPPH activity) under salinity (Figure 1 and Figure 2). Bearing in mind that MDA corresponds with the level of the lipid peroxidation, it could be assumed that the salinity induces changes in the fatty acid of the lipids, leading to alterations in the membrane organization. This statement aligns with previous observations, showing a modification of the fatty acids and membrane fluidity [54]. The experimental results in this study demonstrated a more substantial increase in the amount of H_2_O_2_ in pea (299%) than in maize (110%), which corresponds with a larger amount of MDA in pea compared to maize (Figure 1). These results suggest distinct influences on the processes of lipid peroxidation in both studied species. Specifically, there appears to be greater membrane damage in pea compared to maize. (Figure 4). One of the reasons for better protection in maize under high salinity was the strong increase in flavonoids (Figure 3), which are effective antioxidants [55,56]; a strong increase in their level corresponds with better protection of the functions of the photosynthetic apparatus [35].

Salt treatment also influenced the energy transfer between the complexes of the thylakoid membranes (Table 2). The ratio F_735_/F_685_ increased after treatment with the highest NaCl concentration in both studied species, indicating an increased energy transfer from PSII to PSI. The influence on the energy redistribution between both photosystems could be due to increased lateral mobility of the LHCII as a result of the salt-induced changes in thylakoid membranes. This statement aligns with previous observations for increased PSI antenna size and uncoupling of thylakoid membranes under salinity [57,58,59]. A similar influence of the salinity on the energy transfer between both photosystems was registered in wheat and *Paulownia* [26,27]. Concurrently, an increased fluorescence was emitted from the PSI antenna in both studied species after applying higher NaCl concentrations (Table 3). Changes in the organization of the PSII complex [43,60,61,62] influence the energy transfer among the pigments within this complex. The F_685_/F_695_ ratio decreased in pea plants after applying 150 mM and 200 mM NaCl (Table 2). These changes correlated with an increase of the amount of LHCII^A^ (Table 3) and non-photochemical quenching in this species under salinity [36]. Increased LHCII aggregation has been observed under heat stress and excessive illumination [63,64] and is suggested to be a defense mechanism against abiotic stress.

The salt-induced changes in the PSII organization [65,66,67], lipid composition [68,69], membrane injury (Figure 4), and energy transfer between pigment–protein complexes of the photosynthetic apparatus (Table 2) influenced the PSII photochemistry in the two studied plant species differently (Figure 5). The electron transport mediated by PSII (H_2_O → BQ) after treatment with 150 mM NaCl and 200 mM NaCl was more strongly inhibited in pea than in maize, which corresponds with the different decreases of the open reaction centers in these species [36]. For a more detailed study of the impact of NaCl on the acceptor side of PSII, we studied Q_A_-reoxidation. The decay of the flash-induced variable fluorescence could be fitted by two components (A_1_ fast and A_2_ slow), characterizing two different pathways of Q_A_-reoxidation: by plastoquinone (PQ) and by recombination of Q_A_Q_B_^−^ with oxidized S_2_ (or S_3_) of the OEC [70]. Salinity influenced the ratio (A_1_/A_2_) of these processes in pea and in maize differently (Figure 6). Component A_1_, characterizing the interaction with PQ, increased in maize and decreased in pea under salinity (Appendix A). This could result from different sizes of the PQ pool as well as variations of the impact of NaCl on its size in both species [36]. Data also revealed that the component A_2_, characterizing the recombination of electrons in Q_A_Q_B_^−^ via the Q_A_^−^Q_B_ ↔ Q_A_Q_B_^−^ charge equilibrium with oxidized S_2_ (or S_3_) of the OEC, decreases in maize and increases in pea. It could be concluded that salinity influenced the two pathways of Q_A_ reoxidation differently.

High salinity inhibited the oxygen evolution under both continuous and flash illumination (Figure 7 and Figure 8). The amplitude of the oxygen burst under continuous illumination (A) corresponded with all functionally active PSII centers (PSIIα and PSIIβ) [41]. The curves under continuous illumination in all studied variants exhibit biphasic exponential decay (fast component A_F_ and slow component A_S_), and their ratio A_F_/A_S_ corresponds with the ratio of the functionally active PSIIα to PSIIβ centers [41]. The impact of salinity on this ratio was observed only in pea. This ratio decreased due to stronger salt-induced changes in PSIIα centers, which correspond with a stronger influence on the flash oxygen evolution (Y_3_) than the oxygen evolution under continuous illumination (A) (Figure 7 and Figure 8). The kinetic parameters of the oxygen evolution under high salinity suggest a modification of Mn clusters (Table 4). According to the model of Kok, for the production of one molecule of oxygen, OEC passes through five states (S_0_–S_4_) in same PSII reaction center. The dark-adapted thylakoid membranes contain more stable S_0_ and S_1_ sates [71]. The salt treatment led to an increase of the PSII centers in S_0_ states (Mn^2+^, Mn^3+^, Mn^4+^, Mn^4+^), which is lower by one oxidizing equivalent than S_1_ (Mn^3+^, Mn^3+^, Mn^4+^, Mn^4+^). This fact reveals the influence of the S_0_–S_1_ state distribution of PSII (after salt treatment). At the same time, an increase in misses (α), double hits (β), and blocked PSII reaction centers (S_B_) were registered, while the rate constant of turnover of PSII reaction centers (K_D_) decreased under salt stress. The effects on these parameters were more pronounced in pea than in maize (Table 4), which corresponded with stronger inhibition of PSII in this plant species.

## 4. Materials and Methods

### 4.1. Plant Materials and Treatment

In this study, maize (*Zea mays* L. Method) and pea (*Pisum sativum* L. Ran1) plants were used. The maize seeds were obtained from Euralis Ltd. (Lescar, France), and the pea seeds were sourced from Agrogradina.bg (https://www.agrogradina.bg/semena-grah-ran-1 (accessed on 2 March 2022). The plants were cultivated in a growth chamber under controlled conditions: 28 °C (daytime)/23 °C (nighttime), 150 µmol photons/m^2^ s light intensity, a 12 h light/dark period, and 60% humidity. They were grown in a half-strength Hoagland solution. After 14 days of growth, the plants were treated with varying NaCl concentrations of 50 mM, 150 mM, and 200 mM. We evaluated the impact of NaCl after 5 days.

### 4.2. Isolation of Thylakoid Membranes

Thylakoid membranes were isolated from pea as described in [52] and from maize following the protocol described in [72]. The isolated membranes were resuspended in a buffer solution containing 40 mM Hepes (pH 7.6), 10 mM NaCl, 5 mM MgCl_2_, and 400 mM sucrose. The Chl content in thylakoid membranes was determined as described in [73].

### 4.3. Pigment Content in Leaves

The amount of the pigments in leaves were determined as described in [74]. The amounts of Chl *a*, Chl *b*, and Car were measured using a Specord 210 PLUS spectrophotometer (Edition 2010, Analytik-Jena AG, Jena, Germany), and the pigment content was calculated using Lichtenthaler’s equations [73]. The pigment amount was calculated as mg per g of dry weight (DW).

### 4.4. Membrane Injury Index

The membrane injury index (MII) was assessed as described in [75]. Leaf segments were placed in distilled water for 24 h at room temperature. Afterward, the electrical conductivity of the solutions was determined using a conductometer (Hydromat LM302, Witten, Germany). Following this, the samples were boiled for 30 min and then cooled to room temperature for the determination of the electrical conductivity. The injury index values were determined by the equation: MII (%) = [1 − (1 − T_1_/T_2_)/(1 − C_1_/C_2_)] × 100, where T_1_ and T_2_ are the electrical conductivity of treated samples before and after boiling, respectively, and C_1_ and C_2_ are the values for the untreated control samples [75].

### 4.5. Oxidative Stress Markers and Flavonoids

The levels of lipid peroxidation were measured by determining the malondialdehyde (MDA) content, following the method described in [76]. The content of MDA was determined by measuring the absorbance at 532 nm (Specord 210 Plus, Edition 2010; Analytik Jena AG, Germany) and applying the molar extinction coefficient of 0.155 µM^−1^ cm^−1^. The amount of H_2_O_2_ was measured spectrophotometrically at 390 nm (Specord 210 Plus, Edition 2010; Analytik Jena AG, Jena, Germany) as described in [77]. The molar extinction coefficient of 0.28 µM^−1^ cm^−1^ was used. The results were expressed in nmoles per g of DW.

The total flavonoid content was assessed as described in Stefanov et al. [35]. The absorption at 510 nm was measured using a Specord 210 Plus spectrophotometer (Edition 2010, Analytik Jena AG, Germany). The determination of flavonoid content utilized rutin as a standard, and the total flavonoids present in the plant extract were quantified and expressed as mg of rutin equivalent per g of DW.

### 4.6. Free Radical Scavenging Activity Assay and Ferric-Reducing Antioxidant Power Assay

The total free radical potential of leaf methanol extracts from different pea and maize treatments was assessed using the DPPH^•^ (2,2-diphenyl-1-picrilhydrazil radical) as described in [35]. The absorption was measured at 515 nm on the Specord 210 Plus spectrophotometer (Edition 2010, Analytik Jena AG, Germany).

The ferric-reducing antioxidant power assay (FRAP method) is based on the reduction of a ferric-tripyridyl triazine complex to its ferrous-colored form in the presence of antioxidants. The method is used for the determination of the total antioxidant capacity. The FRAP analysis was made as described in [35]. The samples were measured at 593 nm, and the antioxidant potential of the extracts was determined from a standard curve expressed as μmol Fe^2+^ per g of DW.

### 4.7. Chlorophyll Fluorescence Measurements

The chlorophyll fluorescence emission spectra at low temperature (77 K) were measured using a Jobin Yvon (JY3) spectrofluorometer equipped with a liquid-nitrogen device. The isolated thylakoid membranes were suspended in a solution consisting of 40 mM HEPES (pH 7.6), 10 mM NaCl, 5 mM MgCl_2_, and 400 mM sucrose. The chlorophyll concentration was 20 μg Chl ml^-^. The samples were quickly frozen in a cylindrical quartz cuvette by plunging into liquid nitrogen. The chlorophyll fluorescence emission spectra were recorded from 650 nm to 780 nm, with a slit width of 4 nm. The chlorophyll fluorescence was excited at 436 nm. The chlorophyll emission ratios F_735_/F_685_, used to assess energy redistribution between the two photosystems, and F_685_/F_695_, indicating energy transfer between chlorophyll protein complexes in the PSII complex, were evaluated. Gaussian decomposition of the fluorescence emission spectra was performed following the method described in [38].

The chlorophyll *a* fluorescence after excitation by a saturated light pulse (3000 µmol photons/m^2^s) in dark-adapted leaves was measured using a PAM fluorometer (model 101/103, Walz GmbH, Effeltrich, Germany). The decay components A_1_ and A_2_ of the variable fluorescence relaxation and their times (t_1_) and (t_2_) were determined.

### 4.8. Photochemical Activity of PSII

The photochemical activity of PSII (PSII-mediated electron transport) in isolated thylakoid membranes was measured using an oxygen Clark-type electrode (Hansatech DW1). The measurements were made in a temperature-controlled cuvette under saturating white light intensity at room temperature. The photochemical activity of PSII was assessed in the presence of 0.4 mM exogenous electron acceptor BQ in a reaction medium: 20 mM MES (pH 6.5), 400 mM sucrose, 5 mM MgCl_2_, 10 mM NaCl, and 25 µg Chl/mL.

### 4.9. Oxygen Evolution Measurements

Flash-induced oxygen yields and initial oxygen evolution under continuous illumination of isolated thylakoid membranes were determined by a polarographic oxygen electrode (Joliot-type), as described in Zeinalov [78]. The chlorophyll concentration was 200 μg Chl/mL. The reaction medium contained 40 mM HEPES (pH 7.6), 400 mM sucrose, 10 mM NaCl, and 5 mM MgCl_2_. Oxygen flash yields were induced by periodic flash light sequences, as described in [79]. The initial S_0_ state in darkness, misses (α), and double hits (β) were assessed through fitting least-square deviations to the theoretically calculated yields based on Kok’s model, utilizing the experimentally received oxygen flash yields [80]. The parameters S_B_ and K_D_ were derived through an expanded kinetic adaptation of Kok’s model [81], relying on measurements of the flash spacing variation (1.0 s, 0.7 s, and 0.5 s). The rate constants (k_F_ and k_S_) representing the fast and slow PSII oxygen evolution, characterizing the initial oxygen burst under continuous illumination, were determined by fitting the decay curve of the oxygen burst with two exponential components, as described in [41].

### 4.10. Statistics

Mean values (±SE) were calculated from eight replicates per variant. Two-way ANOVA was used to identify significant differences (*p* < 0.05).

Correlations between chlorophyll fluorescence emission ratios at low temperatures for maize and pea under both control and salt conditions were investigated. Correlations among various rate constants and decay kinetics were also analyzed. Principal Component Analysis was conducted on four plant variants using the correlation matrix of average values after auto scaling. Statistical analysis was performed using Origin 9.0 (Origin(Pro), "Version 9.0.0 SR2" released December 2012, OriginLab Corporation, Northampton, MA, USA.), with Pearson coefficients used for linear correlations. Each data point corresponds to the average value from eight replicates, with significance determined at *p* < 0.05. PCA variable contributions can be found in Appendix A.

## 5. Conclusions

In summary, the experimental results provide new detailed insights into the impact of salinity on the function of the donor and acceptor sides of the PSII complex in species with different salt tolerance. Higher NaCl concentrations (150 mM and 200 mM) inhibited oxygen evolution, a result of modifications to the Mn clusters, which influenced the kinetic parameters of the oxygen-evolving reactions. The effects on the state distribution (S_0_-S_1_), the increase in misses (α), double hits (β), and blocked PSII reaction centers (S_B_), and a decrease in the rate constant of turnover of PSII reaction centers (K_D_) were more pronounced in pea than in maize. Simultaneously, the influence on the pathways of Q_A_-reoxidation varied in both species. In maize, the dominant process was related to the interaction between Q_A_ and PQ, while in pea, the electron recombination of Q_A_Q_B_- with oxidized S_2_ (or S_3_) of the OEC was more pronounced. These changes in the PSII were linked to an influence on the energy transfer between pigment–protein complexes, a decrease in pigment content, and an increase in oxidative stress markers. This study revealed some of the reasons for the difference in salt tolerance of pea and maize.

## Figures and Tables

**Figure 1 plants-13-01025-f001:**
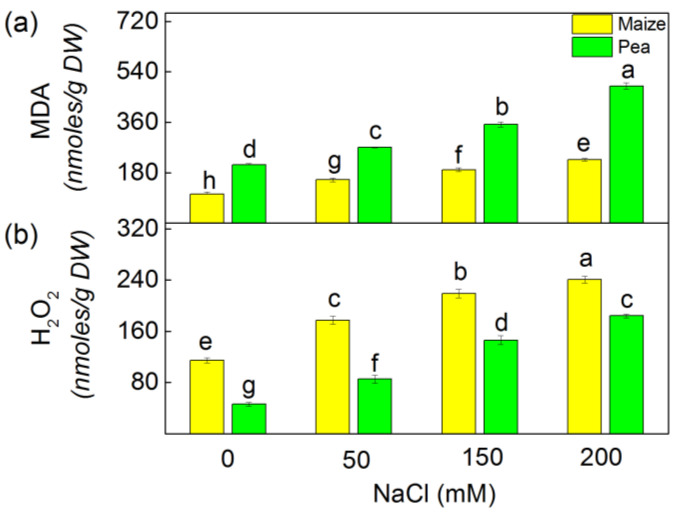
The amounts of MDA (**a**) and H_2_O_2_ (**b**) in maize (*Zea mays* L. Method) and in pea *(Pisum sativum* L. Ran 1) after NaCl treatment for 5 days. Mean values (±SE) were calculated from 8 independent measurements. Different letters indicate significant differences among variants for respective parameters at *p* < 0.05.

**Figure 2 plants-13-01025-f002:**
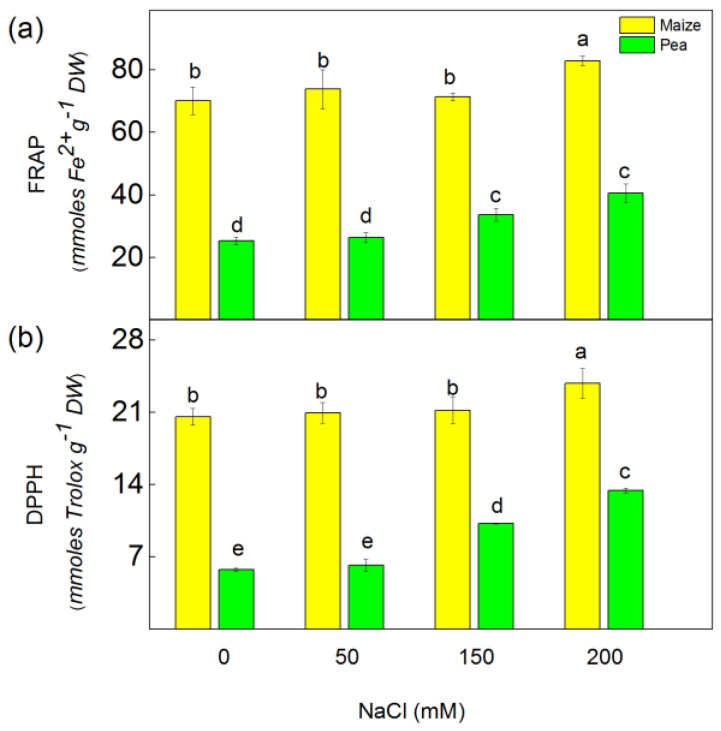
Total antioxidant activity (FRAP) (**a**) and free radical scavenging activity (DPPH) (**b**) in maize (*Zea mays* L. Method) and in pea (*Pisum sativum* L. Ran 1) after NaCl treatment for 5 days. Mean values (±SE) were calculated from 8 independent measurements. Different letters indicate significant differences among variants for respective parameters at *p* < 0.05.

**Figure 3 plants-13-01025-f003:**
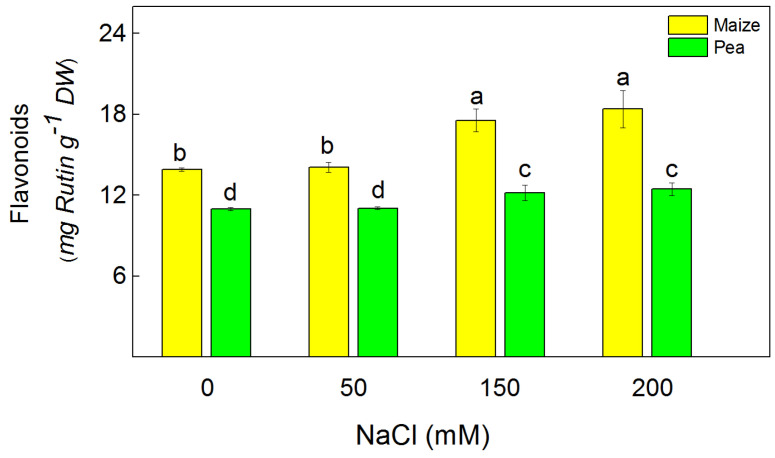
The total flavonoid content in maize (*Zea mays* L. Method) and in pea (*Pisum sativum* L. Ran 1) after NaCl treatment for 5 days. Mean values (±SE) were calculated from 8 independent measurements. Different letters indicate significant differences among variants at *p* < 0.05.

**Figure 4 plants-13-01025-f004:**
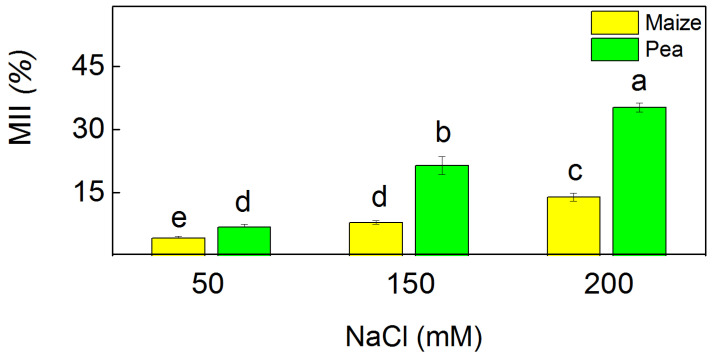
Membrane injury index (MII) in maize (*Zea mays* L. Method) and in pea (*Pisum sativum* L. Ran 1) after NaCl treatment for 5 days. Mean values (±SE) were calculated from 8 independent measurements. Different letters indicate significant differences among variants at *p* < 0.05.

**Figure 5 plants-13-01025-f005:**
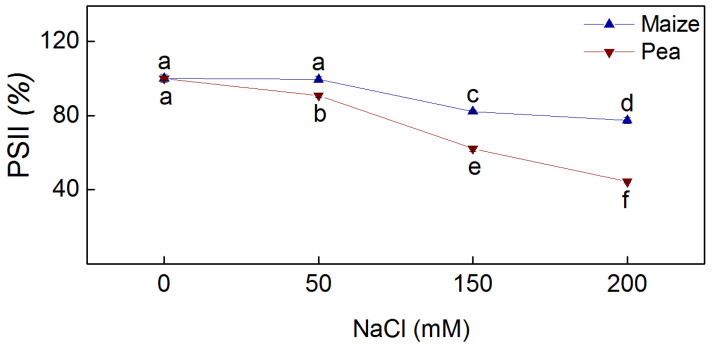
Photochemical activity of PSII (H_2_O → BQ) of isolated thylakoid membranes from leaves of maize (*Zea mays* L. Method) and pea (*Pisum sativum* L. Ran 1) after 5 days of NaCl exposure. The values are expressed as a percentage of the respective control. Different letters denote significant changes between variants at *p* < 0.05.

**Figure 6 plants-13-01025-f006:**
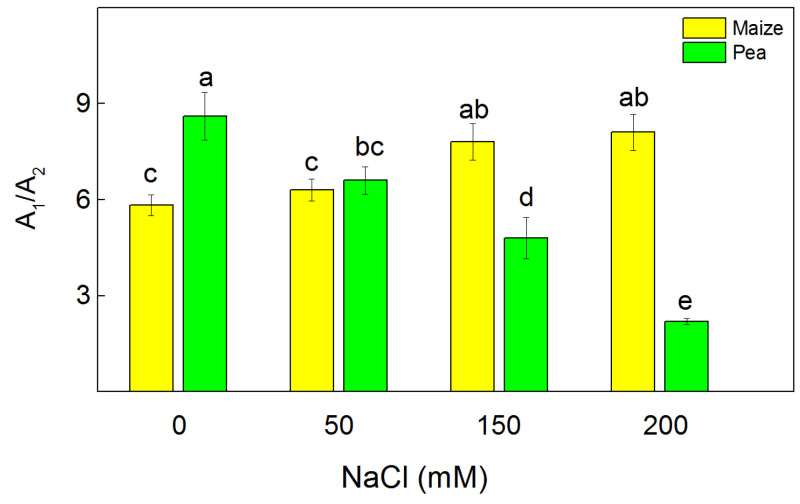
The influence of the different NaCl concentrations on the ratio of the amplitudes of the fast and the slow component (A_1_/A_2_) of the dark relaxation of chlorophyll fluorescence excited by a single saturating light in leaves of maize (*Zea mays* L. Method) and pea (*Pisum sativum* L. Ran 1). Mean values (±SE) were calculated from 8 independent measurements. Different letters indicate significant differences among variants at *p* < 0.05.

**Figure 7 plants-13-01025-f007:**
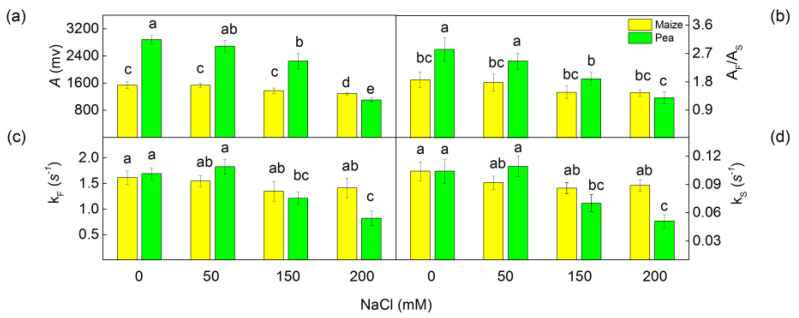
The influence of the different NaCl concentrations of the oxygen evolution under continuous illumination of isolated thylakoid membranes from leaves of maize (*Zea mays* L. Method) and pea (*Pisum sativum* L. Ran 1) after NaCl treatment for 5 days: (**a**) the amplitude of oxygen evolution under continuous illumination; (**b**) the ratio of fast to slow components (A_F_/A_S_); (**c**,**d**) the rate constants (k_F_, k_S_) of oxygen evolution under continuous illumination. Mean values (±SE) were calculated from 8 independent measurements. Different letters indicate significant differences among variants for respective parameters at *p* < 0.05.

**Figure 8 plants-13-01025-f008:**
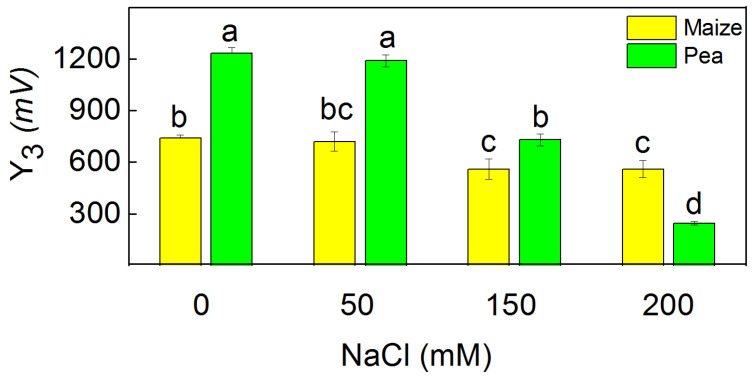
The flash oxygen yield (Y_3_) of isolated thylakoid membranes from leaves of maize (*Zea mays* L. Method) and pea (*Pisum sativum* L. Ran 1) after NaCl treatment for 5 days. Mean values (±SE) were calculated from 8 independent measurements. Different letters indicate significant differences among variants at *p* < 0.05.

**Table 1 plants-13-01025-t001:** The amounts of leaf total chlorophyll (Chl) and carotenoid (Car) content and the pigment ratios Chl *a*/*b* and Car/Chl in maize (*Zea mays* L. Method) and in pea (*Pisum sativum* L. Ran 1) after NaCl treatment for five days. Mean values (±SE) were calculated from 8 independent measurements. Different letters indicate significant differences between the values in the same column at *p* < 0.05.

NaCl (mM)	Chl (mg/g DW)	Car (mg/g DW)	Chl *a*/*b*	Car/Chl
*Zea mays* L.				
0	29.96 ±.1.71 ^a^	5.82 ±.0.30 ^a^	4.65 ± 0.12 ^a^	0.195 ± 0.002 ^c^
50	29.98 ±.1.38 ^a^	5.83 ±.0.21 ^a^	4.46 ± 0.06 ^a^	0.195 ± 0.003 ^c^
150	25.80 ±.1.94 ^b^	5.15 ±.0.34 ^bc^	4.43 ± 0.06 ^a^	0.201 ± 0.003 ^c^
200	17.75 ±.1.05 ^c^	3.97 ±.0.24 ^d^	4.64 ± 0.10 ^a^	0.224 ± 0.002 ^b^
*Pisum sativum* L.				
0	26.46 ±.2.37 ^ab^	6.14 ±.0.25 ^a^	3.22 ± 0.15 ^d^	0.234 ± 0.013 ^b^
50	24.26 ±.0.64 ^b^	5.23 ±.0.16 ^a^	3.56 ± 0.05 ^c^	0.216 ± 0.009 ^b^
150	18.86 ±.1.49 ^c^	4.57 ±.0.37 ^cd^	3.88 ± 0.20 ^b^	0.242 ± 0.009 ^b^
200	11.06 ±.1.12 ^d^	2.95 ±.0.29 ^e^	4.02 ± 0.11 ^b^	0.267 ± 0.002 ^a^

**Table 2 plants-13-01025-t002:** Low-temperature (77 K) fluorescence emission ratios F_735_/F_685_ and F_685_/F_695_ of isolated thylakoid membranes from leaves of maize (*Zea mays* L. Method) and pea (*Pisum sativum* L. Ran 1) after NaCl treatment for 5 days. The thylakoid membranes were excited with 436 nm. Mean values (±SE) were calculated from 8 independent measurements. Different letters indicate significant differences between the values in the same column at *p* < 0.05.

NaCl (mM)	F_735_/F_685_	F_685_/F_695_
*Zea mays* L.		
0	1.50 ± 0.16 ^c^	1.19 ± 0.05 ^a^
50	1.58 ± 0.06 ^c^	1.17 ± 0.02 ^a^
150	1.52 ± 0.11 ^c^	1.15 ± 0.07 ^a^
200	1.85 ± 0.06 ^b^	1.13 ± 0.09 ^a^
*Pisum sativum* L.		
0	1.46 ± 0.15 ^c^	1.13 ± 0.10 ^a^
50	1.52 ± 0.06 ^c^	1.12 ± 0.04 ^a^
150	1.81 ± 0.08 ^b^	0.94 ± 0.11 ^b^
200	2.06 ± 0.09 ^a^	0.95 ± 0.08 ^b^

**Table 3 plants-13-01025-t003:** Fluorescence emission from the pigment–protein complexes in thylakoid membranes of maize (*Zea mays* L. Method) and pea (*Pisum sativum* L. Ran 1) treated for 5 days with NaCl: fluorescence emitted from monomers and trimers of LHCII (LHCII^M+T^), PSII reaction center (PSII RC), PSII antenna, aggregated LHCII (LHCII^A^), PSI core, and PSI antenna. The thylakoid membranes were excited with 436 nm. The area was calculated as % from the total area of emission spectra. Mean values (±SE) were calculated from 8 independent measurements. Different letters indicate significant differences between the values in the same column at *p* < 0.05.

NaCl (mM)	Area (%)					
	LHCII ^M+T^	PSII RC	PSII Antenna	LHCII ^A^	PSI Core	PSI Antenna
*Zea mays* L.						
0	7.69 ± 0.50 ^b^	17.10 ± 0.47 ^a^	15.00 ± 0.66 ^bc^	14.60 ± 0.96 ^cd^	20.70 ± 1.11 ^ab^	24.91 ± 1.01 ^c^
50	8.21 ± 0.39 ^b^	16.58 ± 0.63 ^ab^	14.27 ± 1.10 ^bc^	16.18 ± 0.35 ^bc^	20.84 ± 0.91 ^ab^	23.92 ± 0.46 ^c^
150	8.47 ± 0.62 ^b^	15.42 ± 0.95 ^b^	15.34 ± 0.73 ^bc^	15.16 ± 0.72 ^cd^	21.05 ± 1.07 ^ab^	24.57 ± 1.07 ^c^
200	6.62 ± 0.26 ^c^	15.50 ± 0.50 ^b^	13.85 ± 0.49 ^c^	13.52 ± 0.71 ^d^	24.05 ± 1.54 ^a^	26.45 ± 0.77 ^b^
*Pisum sativum* L.						
0	9.93 ± 0.27 ^a^	16.68 ± 0.65 ^ab^	14.80 ± 0.72 ^bc^	17.22 ± 0.35 ^b^	19.75 ± 1.64 ^ab^	21.62 ± 1.37 ^c^
50	10.61± 0.66 ^a^	16.94 ± 0.89 ^ab^	14.20 ± 1.12 ^bc^	17.24 ± 0.39 ^b^	19.76 ± 1.51 ^ab^	21.25 ± 1.60 ^c^
150	4.85 ± 0.25 ^d^	12.23 ± 0.59 ^c^	16.16 ± 0.82 ^ab^	18.36 ± 0.40 ^a^	19.24 ± 0.80 ^b^	29.16 ± 0.40 ^a^
200	3.99 ± 0.25 ^e^	10.69 ± 0.46 ^d^	18.44 ± 0.53 ^a^	18.44 ± 0.48 ^a^	19.59 ± 1.26 ^b^	28.84 ± 0.63 ^a^

**Table 4 plants-13-01025-t004:** Kinetic parameters of oxygen evolution of isolated thylakoid membranes from leaves of maize (*Zea mays* L. Method) and pea (*Pisum sativum* L. Ran 1) after NaCl treatment for five days: S_0_—PSII centers in the S_0_ state in the dark; α—misses; β—double hits; S_B_—blocked PSII reaction centers; and K_D_—rate constant of turnover of PSII reaction centers. Mean values (±SE) were calculated from 8 independent measurements. Different letters indicate significant differences between the value in the same column at *p* < 0.05.

NaCl (mM)	So (%)	α (%)	β (%)	S_B_ (a.u.)	K_D_ (*s*^−1^)
*Zea mays* L.					
0	21.60 ± 0.54 ^d^	31.54 ± 0.28 ^b^	4.10 ± 0.28 ^bc^	1.53 ± 0.03 ^c^	3.54 ± 0.04 ^a^
50	21.60 ± 0.30 ^d^	30.22 ± 1.29 ^b^	4.42 ± 0.05 ^b^	1.68 ± 0.13 ^bc^	3.75 ± 0.14 ^a^
150	23.16 ± 1.65 ^cd^	32.01 ± 0.62 ^b^	6.24 ± 0.63 ^a^	1.74 ± 0.11 ^bc^	3.33 ± 0.38 ^ab^
200	28.83 ± 0.12 ^b^	34.05 ± 0.97 ^a^	6.63 ± 0.87 ^a^	2.03 ± 0.07 ^a^	3.29 ± 0.14 ^b^
*Pisum Sativum* L.					
0	19.93 ± 0.11 ^d^	24.32 ± 0.32 ^c^	3.90 ± 0.15 ^c^	1.02 ± 0.14 ^e^	3.08 ± 0.14 ^b^
50	20.77 ± 0.53 ^d^	23.89 ± 1.19 ^c^	4.00 ± 0.09 ^c^	1.30 ± 0.08 ^de^	3.30 ± 0.28 ^ab^
150	25.31 ± 1.45 ^c^	25.41 ± 1.85 ^c^	4.35 ± 0.32 ^bc^	1.48 ± 0.10 ^cd^	2.41 ± 0.22 ^c^
200	45.67 ± 1.47 ^a^	31.40 ± 0.72 ^b^	8.96 ± 1.04 ^a^	2.02 ± 0.10 ^ab^	1.80 ± 0.01 ^d^

## Data Availability

Data are contained within the article and Appendix A.

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
