# Peer review of "Changes in Photosystem II Complex and Physiological Activities in Pea and Maize Plants in Response to Salt Stress"

_plants, 2024, doi:10.3390/plants13071025_

Round 1

Reviewer 1 Report

Comments and Suggestions for Authors

In this study, they investigated the effects of salt stress on photosystem II in pea and maize to reveal the role of photosystem in plant resistance to stress. The changes of photosynthetic activity in pea and maize under different salt concentrations were characterized by measuring pigment content, oxidation markers, antioxidant capacity, membrane damage index, PSII. activity and other parameters. The results showed that high concentration of salt stress effectively inhibited the photochemical activity of PSII., which was mainly reflected in the inhibition of energy transfer between pigment-protein complexes, QA reoxidation and oxygen-evolving complex (OEC), and the effect of high concentration of salt stress on pea was more significant than that of maize.

Some comments:

1. In addition to the effect of salt stress, NaCl also has the influence of osmotic factors. In a similar study, control groups such as isotonic NaCl, KCl, NaN03, KN03 and PEG treatments were set up to exclude this factor, indicating that the effect on photosystem II was due to salt stress.

2. In the introduction section, the introduction and description of the research progress of salt stress need to add important content. The latest important literature related to salt stress should be cited, including van Zelm et al. doi: 10.1146/annurev-arplant-050718-100005; Wang et al. 2023, https://doi.org/10.1111/nph.19188.

3. Results 2.1 does not mention the increase of pigment content in the lower concentration of NaCl treatment, and the pigment content does not have a negative correlation with the increase of NaCl concentration, does this mean that the salt treatment at low concentration of maize is conducive to the synthesis and accumulation of pigments? In addition, the Chl a/b value of pea increased with the increase of salt concentration, but the Chl a/b value of corn decreased first and then increased, which should be noted.

4. Results section table2 "the reaction center of the PSII complex (PSII RC), the primary antenna complex of PSII (PSII antenna), LHCII (aggregated trimers, LHCIIA), core complex of PSI (PSI core), and the antenna complex of PSI (PSI antenna).” It does not conform to the syntax, and the juxtaposition of the last and is not preceded by a semicolon.

5. Results section Figure 7 "(a) the amplitude of oxygen evolution under continuous illumination (; (b) the ratio of fast to slow components (AF/AS); (c,d) the rate constants (kF, kS) of oxygen evolution under continuous illumination.” Wrong symbol.

6. In the Materials and Methods section, note the bibliographic citation format "by Andreeva et al. [36]." “as in Rashkov et. al. [77].”“ as in Ivanova et al., [39].”

Comments on the Quality of English Language

Appropriate modification

Author Response

Report to the comments of reviewer 1 on manuscript plants-2925878, by Stefanov et al.

The authors would like to thank the reviewer for constructive and insightful comments in relation to this work. We considered all comments and suggestions to be justified, and corrected the manuscript accordingly. Please, find the detailed list of all edits below. The newly edited text parts are indicated with red letter.

Comments and Suggestions for Authors

In this study, they investigated the effects of salt stress on photosystem II in pea and maize to reveal the role of photosystem in plant resistance to stress. The changes of photosynthetic activity in pea and maize under different salt concentrations were characterized by measuring pigment content, oxidation markers, antioxidant capacity, membrane damage index, PSII. activity and other parameters. The results showed that high concentration of salt stress effectively inhibited the photochemical activity of PSII., which was mainly reflected in the inhibition of energy transfer between pigment-protein complexes, QA reoxidation and oxygen-evolving complex (OEC), and the effect of high concentration of salt stress on pea was more significant than that of maize.

Some comments:

  1. In addition to the effect of salt stress, NaCl also has the influence of osmotic factors. In a similar study, control groups such as isotonic NaCl, KCl, NaN03, KN03 and PEG treatments were set up to exclude this factor, indicating that the effect on photosystem II was due to salt stress.

Answer: We agree that osmotic factors also have an effect on the PSII, because salinity induced both osmotic and ionic stress, but in this study we accessed the effects of both factors.

  1. In the introduction section, the introduction and description of the research progress of salt stress need to add important content. The latest important literature related to salt stress should be cited, including van Zelm et al. doi: 10.1146/annurev-arplant-050718-100005; Wang et al. 2023, https://doi.org/10.1111/nph.19188.

Answer: Changes have been made to the introduction and suggested authors have been included.

  1. Results 2.1 does not mention the increase of pigment content in the lower concentration of NaCl treatment, and the pigment content does not have a negative correlation with the increase of NaCl concentration, does this mean that the salt treatment at low concentration of maize is conducive to the synthesis and accumulation of pigments? In addition, the Chl a/b value of pea increased with the increase of salt concentration, but the Chl a/b value of corn decreased first and then increased, which should be noted.

Answer: It was found that 50 mM NaCl did not affect the amount of chlorophylls and carotenoids, indicating that the synthesis and degradation of pigments were not affected. The amount of pigments (Chl and Car) in pea decreases slightly, but there are no statistically significant differences. We speak for an increase in the Chl a/b ratio only in pea because there are no statistically significant differences between the ratio values in maize.

  1. Results section table2 "the reaction center of the PSII complex (PSII RC), the primary antenna complex of PSII (PSII antenna), LHCII (aggregated trimers, LHCIIA), core complex of PSI (PSI core), and the antenna complex of PSI (PSI antenna).” It does not conform to the syntax, and the juxtaposition of the last and is not preceded by a semicolon.

Answer: Necessary changes have been made in the revised version of the manuscript

  1. Results section Figure 7 "(a) the amplitude of oxygen evolution under continuous illumination (; (b) the ratio of fast to slow components (AF/AS); (c,d) the rate constants (kF, kS) of oxygen evolution under continuous illumination.” Wrong symbol.

Answer: Necessary changes have been made in the revised version of the manuscript

  1. In the Materials and Methods section, note the bibliographic citation format "by Andreeva et al. [36]." “as in Rashkov et. al. [77].”“ as in Ivanova et al., [39].”

Answer: Necessary changes have been made in the revised version of the manuscript

Comments on the Quality of English Language

Appropriate modification

Answer: Some changes were made to the revised manuscript.

At the suggestion of the other reviewer, the title of the manuscript was changed and the figures were edited.

Sincerely yours,

Dr. Emilia Apostolova

Reviewer 2 Report

Comments and Suggestions for Authors

I really appreciated this manuscript from the authors. But I also have some questions.

I think the authors need to clarify in the abstract why they are studying photosystem II and what is its role in plant photosynthesis? What is its significance?

Secondly, the title needs to be reformulated, because the current title does not en summarize the results of the study. I think 'Changes in photosystem II complex and physiological activities in pea and maize plants in response to salt stress' might be more appropriate.

Third, the initials of the plants inside the figure need to be capitalized.

Fourth, why was 100 mM omitted from the NaCl concentration design, which seems unreasonable?

Finally, why did you set up a low-temperature chlorophyll fluorescence response? Instead of room temperature? 

Comments on the Quality of English Language

I think there are a few errors in the language that need to be corrected.

Author Response

Report to the comments of reviewer 2 on the manuscript plants-2925878.

The authors would like to thank the reviewer for constructive and insightful comments in relation to this work. We considered all comments and suggestions to be justified, and corrected the manuscript accordingly. Please, find the detailed list of all edits below. The newly edited text parts are indicated with red letter.

Top of Form

Comments and Suggestions for Authors

I really appreciated this manuscript from the authors. But I also have some questions.

I think the authors need to clarify in the abstract why they are studying photosystem II and what is its role in plant photosynthesis? What is its significance?

Secondly, the title needs to be reformulated, because the current title does not en summarize the results of the study. I think 'Changes in photosystem II complex and physiological activities in pea and maize plants in response to salt stress' might be more appropriate.

 Answer: Thanks for the suggestion. The title of the manuscript is changed.

Third, the initials of the plants inside the figure need to be capitalized.

Answer: We made suggested changes in all figures in the revised manuscript

Fourth, why was 100 mM omitted from the NaCl concentration design, which seems unreasonable?

Answer: The concentrations of NaCl were chosen based on our previous studies. The electrical conductivity of solutions with 50 mM NaCl, 150 mM NaCl and 200 mM NaCl corresponds to moderately saline, very saline and highly saline, respectively (Ivushkin et al., 2019, https://doi.org/10.1016/j.rse.2019.111260).

Finally, why did you set up a low-temperature chlorophyll fluorescence response? Instead of room temperature? 

Answer: We chose the low temperature fluorescence because the bands of the two photosystems are more clear at 77K. In addition, the 77K chlorophyll fluorescence emission spectra allow to evaluate the changes in the PSII complex and the different forms of LHCII.

I think there are a few errors in the language that need to be corrected.

Answer: Some changes were made to the revised manuscript.

Sincerely yours,

Dr. Emilia Apostolova
